# A Vision-Based Motion Control Framework for Water Quality Monitoring Using an Unmanned Aerial Vehicle

Fotis Panetsos [1,*], Panagiotis Rousseas [1], George Karras [1,2], Charalampos Bechlioulis [1,3]
and Kostas J. Kyriakopoulos [1]

[1] Control Systems Laboratory, School of Mechanical Engineering, National Technical University of Athens,
15780 Athens, Greece; prousseas@mail.ntua.gr (P.R.); gkarras@uth.gr (G.K.); chmpechl@upatras.gr (C.B.);
kkyria@mail.ntua.gr (K.J.K.)
[2] Department of Informatics and Telecommunications, University of Thessaly, 35100 Lamia, Greece
[3] Department of Electrical and Computer Engineering, University of Patras, 26504 Patras, Greece
\* Correspondence: fpanetsos@mail.ntua.gr

**Abstract:** In this paper, we present a vision-aided motion planning and control framework for the
efficient monitoring and surveillance of water surfaces using an Unmanned Aerial Vehicle (UAV).
The ultimate goal of the proposed strategy is to equip the UAV with the necessary autonomy and
decision-making capabilities to support First Responders during emergency water contamination
incidents. Toward this direction, we propose an end-to-end solution, based on which the First
Responder indicates visiting and landing waypoints, while the envisioned strategy is responsible
for the safe and autonomous navigation of the UAV, the refinement of the way-point locations that
maximize the visible water surface area from the onboard camera, as well as the on-site refinement
of the appropriate landing region in harsh environments. More specifically, we develop an efficient
waypoint-tracking motion-planning scheme with guaranteed collision avoidance, a local autonomous
exploration algorithm for refining the way-point location with respect to the areas visible to the
drone's camera, water, a vision-based algorithm for the on-site area selection for feasible landing and
finally, a model predictive motion controller for the landing procedure. The efficacy of the proposed
framework is demonstrated via a set of simulated and experimental scenarios using an octorotor UAV.

**Keywords:** UAV; autonomy

## 1. Introduction

Safeguarding nature and especially the water resources of our planet is nowadays,
more than ever, of utmost importance. Human well-being depends on clean potable water,
ecosystems that sustain agriculture rely heavily on unpolluted water supplies, while clean
oceans are essential for the preservation of marine life. Unfortunately, water contamination
can easily occur either by inevitable natural disasters (e.g., floods or earthquakes) or
human-caused malicious actions (e.g., terrorist attacks or chemical accidents). Therefore,
the development of methods and tools for preventing and dealing with water pollution
threats is essential. Toward this direction, the systematic monitoring and surveillance of
water surfaces is a vital and significant process. This, however, comes with its caveats, since
many areas are difficult or even impossible to access by humans in order to perform water
monitoring and sampling. Furthermore, large areas of water must be timely covered during
a typical surveillance scenario. Lastly, in case of natural disasters or malicious actions,
conditions might be extremely harmful for specialized personnel such as First Responders
to attend due to chemical and pathogen pollution or critical environmental conditions.

It is thus evident that there exists a significant need for increasing the effectiveness,
speed and coverage of water monitoring missions. Toward this goal, this paper proposes
a multi-modal perception and control framework, which is applicable to multi-rotor Un-
manned Aerial Vehicles (UAVs), that provides a complete and autonomous solution for the

execution of a water monitoring mission in realistic conditions. The framework presented herein consists a modular, open-source driven and non platform-specific solution that was thoroughly tested in various outdoor experimental scenarios.

### 1.1. Related Literature

Autonomous aerial vehicles have been widely recognised for their versatile nature, which renders them as the most popular choice for a variety of surveillance and monitoring applications [1,2]. Moreover, recent multi-disciplinary engineering advances in structural design, sensors, actuators and communications enhance the overall autonomy of these platforms. Nowadays, UAVs are equipped with fast and dependable communication technologies [3], enhanced AI-based sensing capabilities [4] and robust motion control schemes (e.g., dynamic window approaches [5], Machine Learning/MPC approaches [6]). In fact, the number of commercially available "ready-to-fly" UAVs demonstrates the maturity of this technology. Furthermore, the open-source philosophy has greatly facilitated the process of setting-up and flying UAVs. Both the low-level [7] and high-level [8] aspects of drone control now have available robust, almost "plug-and-play" solutions, while simulation environments expedite the testing and tuning of the software by enabling sensing and hardware simulations [9]. The software-in-the-loop (SITL) philosophy enables almost seamless transition from simulated to real-world experiments.

A typical autonomous surveillance scenario requires efficient motion planning and control schemes. Most common autopilot frameworks support path following based on data fusion of the on-board navigation sensors (GPS, IMU, altimeter, etc.) and mainly linearized PID motion controllers, which can prove quite effective in high altitude and obstacle-free workspaces. However, in more complex missions, e.g., search and rescue or sampling for possible contamination, the vehicle must operate in low altitudes and possibly in cluttered and unknown environments; hence, more sophisticated control schemes are required. Recent improvements on embedded processing units have enabled the adoption of advanced motion planners in UAVs. In fact, some of the most common planning algorithms for known workspaces, such as RRT* (Rapidly-exploring random trees) [10], as well as for unknown environments, such as A* [11] and Dijkstra's [12] algorithm, have been successfully employed in UAV operation. However, these schemes are usually not incorporated in the standard autopilot control system of the UAV and often require specialized personnel to operate the vehicle.

Furthermore, the detailed perception of the environment is also an important aspect in aerial surveillance applications. More specifically, in water surveillance tasks, two equally important issues arise: namely, the inference of three-dimensional information pertaining to the topology of a ground surface and the detection of land and water areas. In general, three-dimensional information can be extracted by incorporating either LiDAR or RGB-D sensors in an UAV's sensor suite [13], while relatively to water monitoring, photogrammetry has been extensively used to obtain 3D information [14,15]. When it comes to detecting water surfaces from top-view images of the Earth's surface, the most widely used approaches employ multi-spectral cameras to obtain multiple images of the same area and subsequently apply a thresholding process, which, if tuned correctly can produce a valid classifier of water and land pixels (see, e.g., [16]). It is however necessary to equip the UAV with a multi-spectral camera to the detriment of the cost and overall weight of the platform.

Another important aspect when flying in unknown areas is autonomous landing. This can be a rather tricky undertaking especially in harsh outdoor environments. A plethora of different autonomous landing approaches is discussed in [17]. The existing solutions pertaining to outdoor landing, which we will focus upon, can mainly be classified into "known" [18–21] and "unknown" environments [22]. The landing task can be broken down into two main phases: (1) finding an appropriate landing spot and (2) executing the landing maneuver. In order to execute these steps, both for known and unknown environments, the autonomous platform is equipped with perception sensors relevant to

the task at hand. Notably, during landing operations in known environments, less complex sensing instruments and algorithms (e.g., plain RGB cameras) are usually utilized owing to the existence of pre-determined landmarks [17], which are often employed to not only provide a suitable location for landing but also to aid in controlling the drone (e.g., through visual control) so as to land safely [23]. In contrast, a priori unknown environments are evidently more challenging, as the on-board computing and sensing instruments need to be utilized in order to analyze the environment and make on-the-fly decisions for choosing a landing location as well as controlling the vehicle to safely land in the appropriate spot.

In this work, we mainly focus on the unknown environments case, since they are relevant to water surveillance and monitoring applications in emergency situations, where no prior knowledge of the environment is available. In such cases, the UAV must be equipped with advanced sensors (e.g., LiDAR or RGB-D) to deal with the variance of information. In [24], data from a LiDAR and an RGB camera were fused to provide a depth map, which was post-processed to calculate a proper landing spot. In [25], an integrated solution with Simultaneous Localization and Mapping (SLAM) was employed to achieve both localization and control of the drone while calculating a safe landing area from a 3D representation of the platform's environment. In [26], ground-station sensing instrumentation was incorporated to facilitate perception. It is evident that a variety of solutions for 3D perception of the environment are available in the related literature. However, any selected solution should be computationally efficient to be able to perform well via purely on-board sensing and embedded computing capabilities.

### 1.2. Contribution

The purpose of this work is to provide an end-to-end efficient solution for autonomous monitoring and surveillance of water surfaces using an UAV by integrating various perception, motion planning and control algorithms in a common framework. Our ultimate goal is the provided framework to be utilized by First Responders operating at emergency incidents, without the need of having specialized technical knowledge or piloting skills. In this way, we aim to reduce the danger posed to First Responders in water contamination scenarios by keeping them at long distance from the contaminated areas and dispatching a UAV instead. In this scope, the contribution of our work can be summarized as follows:

- Integration of an efficient motion planning scheme featuring on-line obstacle detection and collision avoidance, with the common open source autopilot and mission planner systems that most commercial UAVs employ.
- Design and development of a novel autonomous exploration algorithm based on on-board robotic vision, which provably maximizes (locally) the visible water to the UAV's camera.
- Design and development of a novel vision-based scheme for the on-line detection of the most suitable landing area for the UAV in unknown outdoor and harsh environments, based on a stereoscopic camera's depth map from which information about the ground's geometry is inferred.
- Formulation and development of a Non-linear Model Predictive Control (NMPC) scheme for the autonomous landing procedure.

Furthermore, since the proposed framework is highly modular, it can be easily modified to tackle a variety of different real-world scenarios that go beyond water monitoring, (e.g., tasks such as plant monitoring) through altering one or more modules either partially or entirely.

### 1.3. Outline

The rest of the paper is organized as follows: Section 2 provides a description of the problem at hand. Section 3 provides background knowledge regarding the UAV motion model and low-level control. Section 4 presents in detail the methodology that synthesizes the proposed solution. Section 5 demonstrates the efficacy of the overall framework via a set of experimental scenarios. Finally, Section 6 concludes the paper.

## 2. Problem Statement

We assume that after an emergency alert concerning a possible water contamination, a team of First Responders is tasked to monitor an a priori selected set of way-points located near a body of water and in the proximity of land (e.g., river, lake, etc.). Due to the possible presence of pathogens, the monitoring mission will be carried out by a UAV. This monitoring step will henceforth be taken "to visually survey" the water in the nearest vicinity of the above way-point. This could also potentially include further actions, such as water sampling at the indicated areas using an appropriate sampling mechanism. However, in this work, we will focus on visual surveillance.

We consider a multi-rotor UAV equipped with a down-looking stereoscopic camera, a forward-looking stereoscopic camera and appropriate navigation sensors (GPS, IMU, altimeter, etc.) along with common autopilot software for low-level control. The mission consists of the following steps:

- The First Responders communicate information in the form of way-points $W = [w_1, w_2, \cdots, w_{N_W}] \in \mathbb{R}^{2 \times N_W}$ to the UAV via a *High-Level Planner Module*. These points correspond to areas of interest for monitoring purposes. The *High-Level Planner Module* coordinates the rest of the UAV modules (see below) in order to assure the accomplishment of the mission goals.

- The UAV should be autonomously navigated toward the aforementioned way-point set $W$. However, the vehicle must have the ability to detect and avoid on-line unknown obstacles that may appear in route (e.g., trees, buildings, etc.), since significant parts of the flight will be carried out at low altitudes (below 50 m). This task will be handled by the *Motion Planning Module*.

- Since each way-point could likely be located close to land and water, i.e., in case of a river, the down-looking camera is highly possible to include both water and ground areas in the field of view. Hence, a local autonomous exploration algorithm, starting from way-point $w_i$, will guide the UAV accordingly in order to ensure the maximization of visible water surface inside the camera's field of view. This task will be handled by the *Autonomous Exploration Module*.

- Upon completion of the way-points visiting, the UAV should be able to land as close as possible to an indicated way-point $w_l \in \mathbb{R}^2$, which corresponds to an environmentally unknown and unprepared for landing area. Thus, a vision-based scheme will handle the on-line detection of the most suitable landing area for the UAV, as close as possible to the indicated landing way-point, based on the approximated ground's height map. Finally, given the selected landing spot, the UAV must perform the landing maneuver. During the landing phase, the Landing Area Detection algorithm will still be running to provide on-line possible refinement of the landing point. An NMPC motion controller will be responsible for realizing the landing procedure. This task will be handled by the *Autonomous Landing Module*.

An overview of the proposed framework and the related modules is depicted in Figure 1.

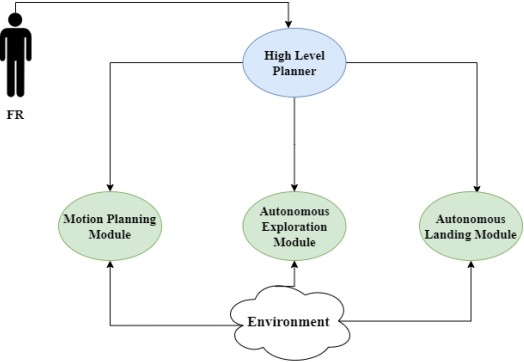

**Figure 1.** Framework overview.

Each of the participating modules will be analyzed in detail in Section 4.

## 3. Preliminaries

### 3.1. Multirotor Kinematics and Dynamics

In this subsection, the well-known kinematic and dynamic models of multi-rotor platforms are introduced. Consider a multi-rotor robot as depicted in Figure 2.

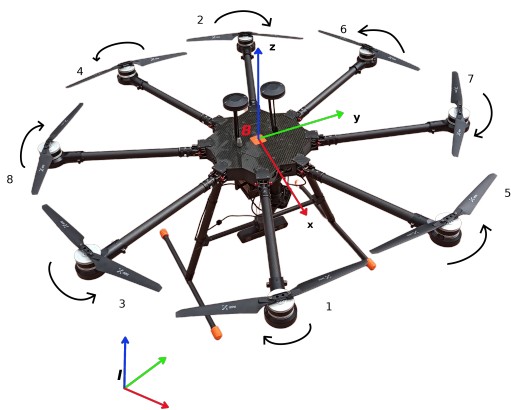

**Figure 2.** Octorotor's frames and motor order.

Let $\mathbf{B} = \left\{ e_{B_x} \quad e_{B_y} \quad e_{B_z} \right\}$ denote the body fixed frame, whose origin coincides with the vehicle's center of mass. In addition, a fixed inertial frame $\mathbf{I} = \left\{ e_{I_x} \quad e_{I_y} \quad e_{I_z} \right\}$ is defined, as shown in Figure 2. The translational and rotational dynamics of the vehicle, while considering external forces and moments, are described by the Newton–Euler equations [27,28]:

$$\dot{\mathbf{p}} = {}^I\mathbf{v} = {}^I\mathbf{R}_B{}^B\mathbf{v} \tag{1}$$

$$m^I\dot{\mathbf{v}} = {}^I\mathbf{R}_B\mathbf{F} \tag{2}$$

$$\mathbf{J}\dot{\boldsymbol{\omega}} = -\boldsymbol{\omega} \times \mathbf{J}\boldsymbol{\omega} + \mathbf{M} \tag{3}$$

where $\mathbf{p} = \begin{bmatrix} p_x & p_y & p_z \end{bmatrix}^T$, ${}^I\mathbf{v} = \begin{bmatrix} v_x & v_y & v_z \end{bmatrix}^T$ are the position and the linear velocity of the multi-rotor with respect to the inertial frame $\mathbf{I}$, ${}^B\mathbf{v} = \begin{bmatrix} u & v & w \end{bmatrix}^T$ is the linear velocity expressed in the body frame $\mathbf{B}$, $m$ is the mass, ${}^I\mathbf{R}_B$ is the rotation matrix used to perform rotations from $\mathbf{B}$ to $\mathbf{I}$, $\mathbf{J}$ is the inertia matrix and $\boldsymbol{\omega}$ is the angular velocity of the vehicle expressed in the body frame $\mathbf{B}$.

The external forces and torques applied to the vehicle's body are divided into:

$$\mathbf{F} = \mathbf{F}_M + \mathbf{F}_d + \mathbf{F}_g \tag{4}$$

$$\mathbf{M} = \mathbf{M}_M + \mathbf{M}_d \tag{5}$$

where:

- $\mathbf{F_d} = C_d{}^B\mathbf{R}_I\|{}^I\mathbf{v}\|{}^I\mathbf{v}$ are the drag forces and $C_d$ is the drag coefficient matrix;
- $\mathbf{F_g} = m^B\mathbf{R}_I\begin{bmatrix} 0 & 0 & -g \end{bmatrix}^T$ is the gravitational force with $g$ denoting the gravitational constant;
- $\mathbf{F_M} = \begin{bmatrix} 0 & 0 & T \end{bmatrix}^T$ is the total thrust generated by the motors;
- $\mathbf{M_M} = \begin{bmatrix} \tau_x & \tau_y & \tau_z \end{bmatrix}^T$ is the torque produced by the motors;
- $\mathbf{M_d} = C_m\|\boldsymbol{\omega}\|\boldsymbol{\omega}$ are the drag moments with $C_m$ denoting the drag moment coefficient matrix.

The total input thrust and moment applied to the multi-rotor is highly dependent on the vehicle's structure and specifically on the number $N$ of motors and the configuration of the airframe. According to momentum theory, thrust force $T_i$ and the drag moment $\tau_i$

produced by the propellers are assumed to be proportional to the square of the rotor's angular velocity, i.e.,

$$T_i = C_T \omega_i^2 \tag{6}$$

$$\tau_i = C_\tau \omega_i^2 \tag{7}$$

where $i = 1, \ldots, N$ and $C_T$, $C_\tau$ are the thrust and drag coefficients correspondingly.

For the octorotor, used in experimental scenarios, the total thrust and moments are computed by the following control allocation matrix:

$$\begin{bmatrix} T \\ \tau_x \\ \tau_y \\ \tau_z \end{bmatrix} = \begin{bmatrix} C_T & C_T & C_T & C_T & C_T & C_T & C_T & C_T \\ -C_T l_x & C_T l_x & -C_T l_x & -C_T l_x & C_T l_x & C_T l_x & C_T l_x & -C_T l_x \\ C_T l_y & -C_T l_y & C_T l_y & -C_T l_y & C_T l_y & -C_T l_y & C_T l_y & -C_T l_y \\ -C_\tau & -C_\tau & C_\tau & C_\tau & C_\tau & C_\tau & -C_\tau & -C_\tau \end{bmatrix} \begin{bmatrix} \omega_1^2 \\ \omega_2^2 \\ \omega_3^2 \\ \omega_4^2 \\ \omega_5^2 \\ \omega_6^2 \\ \omega_7^2 \\ \omega_8^2 \end{bmatrix} \tag{8}$$

with $l_x$, $l_y$ denoting the distance of each motor with respect to the center of mass.

### 3.2. Autopilot and On-Board Sensors

During both the simulation and experimental scenarios, an autopilot, and particularly the open source ArduPilot system [7], is employed in order to provide reliable low-level control of the vehicle. The low-level control is realized by a cascaded PID control structure consisting of an outer position loop and an inner attitude one. More precisely, the outer position loop is responsible for converting the reference position $\mathbf{p}_d$, velocity ${}^I\mathbf{v}_d$ (or ${}^B\mathbf{v}_d$) and heading $\psi_d$ of the vehicle to target orientation (roll $\phi_d$, pitch $\theta_d$, yaw $\psi_d$) and throttle. The inner attitude controller is, then, translating the aforementioned orientation and throttle commands to motor pulse width modulation (PWM) values. The state feedback is achieved by fusing sensor measurements, such as data by GPS, compass and IMU, using an Extended Kalman Filter. An overview of the control architecture is shown in Figure 3.

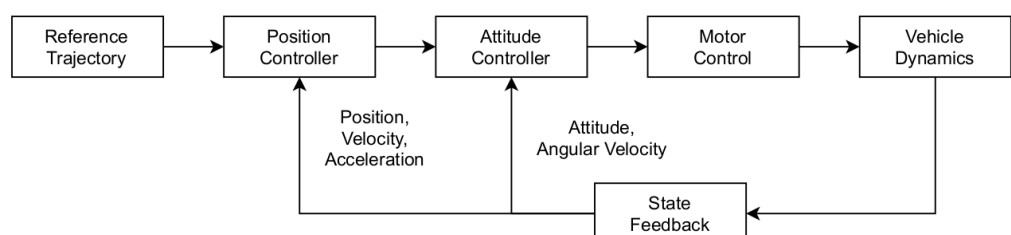

**Figure 3.** Ardupilot control architecture.

### 3.3. Chosen Platfrom Specifications

Finally, in order for the platform to be able to perform in a desirable manner (as will be comprehensively discussed in the later sections), a sensing suite is necessary. For this purpose, the multi-rotor is equipped with two stereoscopic cameras, a forward-looking one, so as to measure the distance to the surrounding obstacles while navigating from one waypoint to another, and a down-looking one in order to both visually detect water surfaces and appropriate landing areas. With regard to the downward-looking camera, the function of a stereoscopic camera is two-fold; first of all, it infers reliable 3D information and secondly, it is used to classify land and water areas. As previously mentioned, while a multi-spectral camera could be used for such a classification, the dual purpose of a single stereoscopic device has the benefits of cost and weight-saving, which are both aspects crucial for the application in question.

## 4. Materials and Methods

In this section, we propose a framework for tackling the above formulated problem of water monitoring via drone platforms. The framework is comprised of three modules, working independently of each other, and managed on a high level by a planner that is in charge of "supervising" the mission and the execution of each module according to a user-provided set of way-points, as previously discussed.

### 4.1. Motion Planning Module

The main objective of this module is the navigation of the vehicle toward the desired locations while, simultaneously, avoiding collisions with obstacles. Therefore, a set of way-points *W* is initially commanded by the pilot using a common Ground Control Station (GCS) and a map of the area of interest. The aforementioned way-points are defined in the World Geodetic System 1984 (WGS84), and thus, a conversion is performed from latitude and longitude coordinates to Cartesian ones in the inertial frame.

The safe navigation of the multi-rotor assumes the existence of a sensor capable of reliably measuring the distance between the vehicle's body and static or dynamic obstacles which jeopardize the success of the mission. Hence, the multi-rotor is equipped with a forward-looking stereoscopic camera and the cumulative depth data are exploited in order to build, in real-time, a 2D costmap of the area, i.e., a 2D occupancy grid where each cell is marked as free, unknown or occupied according to the surrounding obstacles.

Given the costmap, a local planner, specifically the Dynamic Window Approach (DWA) local planner [5], is utilized for producing feasible velocity commands which ensure the avoidance of undesired collisions and the navigation of the vehicle toward the target way-point. The waypoints are sent sequentially to the ROS-based motion planning package, namely the "move_base" package, and the above procedure is repeated until the mission is accomplished.

### 4.2. Autonomous Exploration Module

#### 4.2.1. CNN Water Detection

An important part of the Autonomous Exploration algorithm is the real-time detection and classification of the ground and water surfaces that constitute the environment above which the UAV operates. The platform is equipped with a downward-looking stereoscopic camera. In addition to the 3-dimensional information inferred from this sensor, the RGB images taken in real time can be utilized to extract ground and water pixels.

In order to achieve this classification task, Convolutional Neural Networks were employed. Such Neural Networks have long been used for image analysis and robotic vision related tasks with great success. More specifically, an image segmentation-oriented Neural Network was used to classify pixels as ground or water ones. The following procedure was implemented to prepare the dataset for training:

1. Manual labeling of the images;
2. Binary masks creation from labeled images;
3. Augmentation of the dataset through an open-source software [29];
4. Resizing of the frames from $720 \times 480$ pixels to $128 \times 128$ pixels,
5. Classification for 2 classes (Class 0: Ground and Class 1: Water).

The structure of the proposed CNN (Figure 4) follows: Firstly, the images are passed into the convolutional layer, the normalized output of which is then passed on to the pooling layer. This layer collects data sets from the convolutional layer and samples the output of a result from the selected ones. After a plurality of subsequent convolutional and pooling layers, the final fully connected layers are utilized. The CNN weights are obtained through the back-propagation method. In order to apply the above procedure, the Keras image segmentation framework's vgg_unet CNN was employed [30].

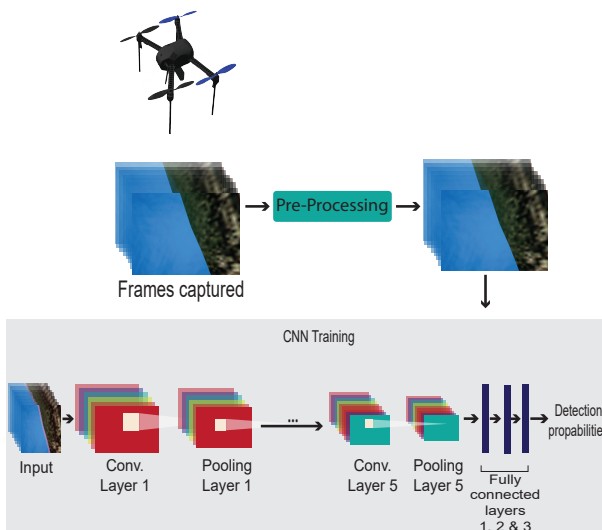

**Figure 4.** CNN architecture for water–land detection.

The CNN was trained over a sample of initially 1500 manually labeled real-world images, which were augmented to a final dataset of 6000 images. The data were gathered in the form of video frames obtained by manually flying the octorotor above aquatic environments. Approximately 10% of the images were used as a validation set, while the rest were utilized for training. The algorithm converged to over 99% accuracy on the test set in 10 training epochs. An example of the output of the trained CNN is depicted in Figure 5.

It is evident that the success rate and the accuracy of the CNN after training is heavily dependent on the quality of the employed data, i.e., the quality of the images, the accuracy of the labels, as well as the surrounding environmental conditions. While we achieved a highly satisfactory performance in the context of the conducted experimental tests (see Figure 5 for an indicative example), augmenting the capabilities of the proposed CNN architecture is still a valid future direction. The aspects upon which performance can be increased mainly concern the generalization capabilities of the CNN to different bodies of water, weather conditions, etc.

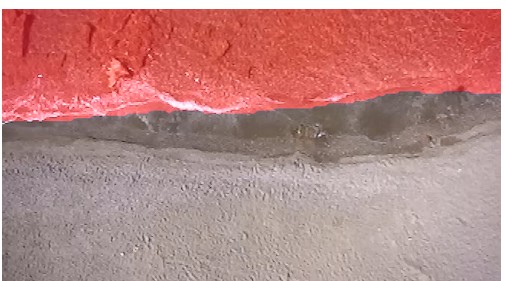 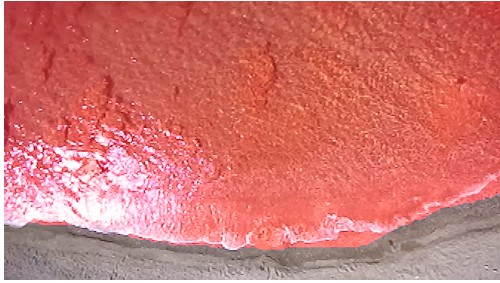

**Figure 5.** Images from the on-board camera combined with the CNN output. The red pixels represent the pixels which the CNN classifies as water pixels. The raw output of the NN consists of a binary mask in which the respective water pixels are highlighted.

### 4.2.2. Visible Water Maximization Algorithm

In order to visually survey an area of a water body, a framework for maximizing the visible (to the drone's downwards-looking camera) water is formulated. Let $\mathcal{I} \subset \mathbb{R}^2$ denote the 2D image plane. Then, this image consists of two parts, namely $\mathcal{I}_w \subseteq \mathcal{I}$ denoting the "water part" of the image and $\mathcal{I}_l \subseteq \mathcal{I}$ denoting the "land part" of the image. Note that

$\mathfrak{I}_w \bigcup \mathfrak{I}_l = \mathfrak{I}$ and $\mathfrak{I}_w \bigcap \mathfrak{I}_l = \varnothing$. We then consider that the drone follows exactly a setpoint velocity control law, i.e., a single integrator model:

$$\dot{p} = {}^I v, \tag{9}$$

where $p = [p_x, p_y]^T \in \mathbb{R}^2$ denotes the drone's longitudinal and lateral position in an inertial frame of reference (state vector), and ${}^I v \in \mathbb{R}^2, {}^I v = [v_x, v_y]^T$ denotes the planar (longitudinal–lateral) velocity control (control–input vector). This assumption is very accurate for relatively small velocities. Along with the fact that normally, the autopilots of relevant multi-rotor aerial vehicles are designed to take such commands as input, while ensuring stability and safety of the vehicle, such an assumption is in practice not only useful but also not limiting in its scope. Furthermore, it is evident that any lateral motion of the drone will be the result of a roll or pitch rotation. The low velocity assumption is also important in this regard, as the following analysis assumes an always downwards-looking camera, which is violated by large rotations in the aforementioned axis. Alternatively, a gimbal can be used to ensure proper orientation of the camera frame. In order to maximize the water that is inside the frame of the camera, we employ the following control law:

$$ {}^I v = s_w(p) \triangleq \frac{1}{A} \left[ S_x(p), S_y(p) \right]^T, \tag{10}$$

where

$$ S_y(p) = \iint_{\mathfrak{I}_w(p)} (y - p_y) dA, \quad S_x(p) = \iint_{\mathfrak{I}_w(p)} (x - p_x) dA, \tag{11}$$

where $A$ denotes the area in the image frame that is occupied by water. The above expression essentially means that we input the position vector of the centroid of the water area $\mathfrak{I}_w$ with respect to the position of the drone. This means intuitively that the drone will tend to move toward the centroid of the water part of the image, which, under mild assumptions, will result in minimizing the area of land that is visible to the drone evidently, thus maximizing the respective area of visible water. We will prove this assertion in Theorem 1.

For the purposes of the following proof, let the water–land boundary be denoted by a function, which is assumed to be an one-to-one mapping from the $x$ to the $y$ coordinates, i.e., $S_p(x) : \mathbb{R} \to \mathbb{R} \in C$ (it is evident how the function of the land–water boundary depends on the position of the drone) expressed with respect to an inertial frame of reference. The proof follows along the same lines even for functions that do not satisfy this assumption by performing strategic cuts on the function on its critical points and breaking up the relevant integrals (this analytical definition can be applied even if the boundary is not a function, i.e. not right–unique, but it can be expressed as such via a rotation of the image plane in the xy plane (see Figure 6)). This process is left out for the sake of brevity; however, it would be imperative to execute such "cuts" in order to properly define the piece-wise inverse functions of the monotonic sections of the respective function for calculating the double integrals of Equation (11). We thus limit ourselves to monotonic ones. One final assumption is that some water is visible in the camera frame upon the execution of the control law (10). The following Theorem 1 proves that the control scheme (10) stabilizes the drone in a position where the visible water to the on-board camera is maximized.

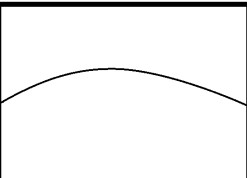 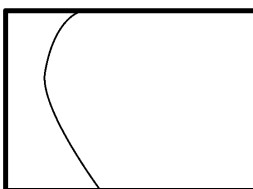

**Figure 6.** Example of a properly defined boundary function (**left**) and an improperly defined one that can be fixed through a simple rotation (**right**).

**Theorem 1.** *Consider a coordinate system centered at the camera frame (see Figure 7), which is assumed to be identical to the drone's position, namely $p \in \mathbb{R}^2$. Then, let $S(x) \triangleq S(x; p) : [-c_u, c_u] \to [-c_v, c_v] \in C^1$ denote a function that describes the land–water boundary with respect to the above frame, where $c_u, c_v$ are the halved dimensions of the image plane in $[x, y]$ respectively (appropriately scaled with respect to the height of the drone to reflect physical, real-world dimensions). We further assume that the characteristics of the shore are such that the body of water is larger than the respective land mass, such that no land encircles a part of water (this assumption is rather mild; the applications at which this framework is targeted would most likely involve large bodies of water, with the drone operating at such heights that even a small lake would satisfy the above assumption. However, if this assumption does not hold, then the drone would still converge with the water body at its center; thus, the monitoring process could still be considered successful). Then, the dynamical system (9) under the control law (10) is asymptotically stable.*

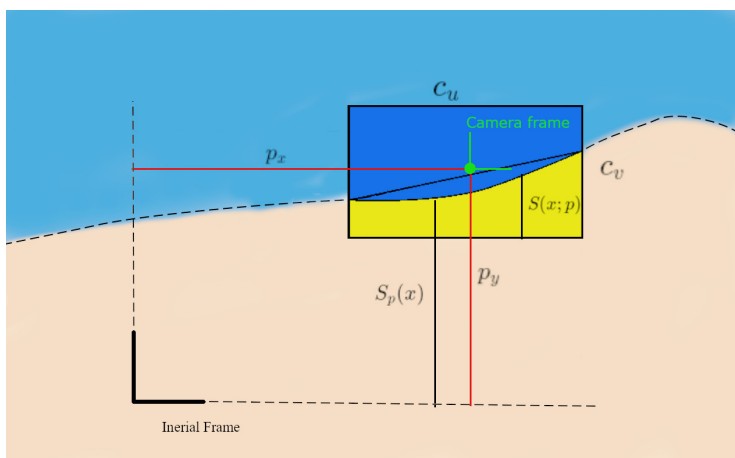

**Figure 7.** Figure of Theorem 1. The relevant frames are depicted along with the function that represents the water–land boundary. The components of the drone's position vector are also depicted.

**Proof.** Consider as a Lyapunov candidate the following function:

$$\mathcal{L}(p) = A(p) = \iint_{\mathcal{I}_l(p)} dA. \tag{12}$$

This function essentially expresses the measure of area of the land part of the ground that the drone observes and is always positive. It is furthermore zero only when the drone observes nothing but water. Thus, it is a valid Lyapunov candidate for the goal in mind, i.e., that the drone only observes water upon convergence. We will now show how the time derivative of the above Lyapunov candidate is negative, except for point(s) of convergence. We have:

$$\begin{aligned} \dot{\mathcal{L}} &= \nabla_p A^T \dot{p} \\ &= \nabla_p A^T s_w(p). \end{aligned} \tag{13}$$

We will prove that

$$\dot{\mathcal{L}} < 0, \tag{14}$$

for points that do not result in the drone observing solely water by showing that the vectors $\nabla_p A$ and $s_w(p)$ are pointing toward opposite directions. Firstly, note that the centroid of the land area, denoted by $s_l(p)$ is contradirectional to $s_w(p)$, owing to the centroid of the whole image being at the origin of the chosen frame of reference and the fact that the two areas combined form the image. To see this, note that given the centroids of two shapes

$s_1, s_2 \in \mathbb{R}^2$ in a given coordinate frame, then the centroid $s_{1,2}$ of the shape resulting from combining the two former ones is given by:

$$s_{1,2} = \frac{A_1 s_1 + A_2 s_2}{A_{1,2}}, \tag{15}$$

where $A_1$, $A_2$, $A_{1,2}$ denote the areas of the respective shapes. Since in our case, the coordinate frame is centered at the center of the camera frame, which coincides with the centroid of the rectangular image, then:

$$s_{l,w} = \frac{A_w s_w(p) + A_l s_l(p)}{A_{l,w}} = \vec{0} \Rightarrow$$
$$s_w(p) = -\frac{A_l}{A_w} s_l(p), \tag{16}$$

which shows that the two centroids are contradirectional (see Figure 8).

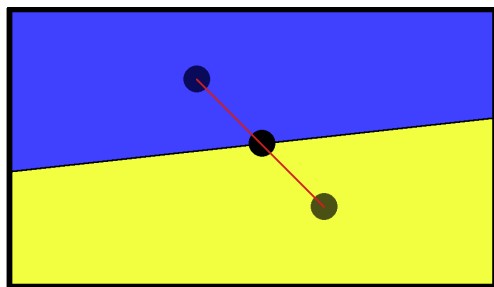

**Figure 8.** The centroids of the water–land areas.

We now only have to prove that $\nabla_p A$ and $s_l(p)$ are codirectional. Assume that the function $S(x)$ is monotonically increasing (see Figure 7). The proof for a monotonically decreasing function follows the same procedure. We have:

$$A(p) = \iint_{\mathcal{I}_l} dA = \int_{p_x - c_u}^{p_x + c_u} S_p(x) - (p_y - c_v)dx, \tag{17}$$

where $S_p(x)$ is the function $S(x)$ expressed with respect to the inertial frame of reference. Thus:

$$\nabla_p A = [S_p(p_x + c_u) - S_p(p_x - c_u), -2c_u]^T = [S(c_u) - S(-c_u), -2c_u]^T. \tag{18}$$

Furthermore, expressed at the new frame of reference (camera frame center), the coordinates of $s_l(p) = \left[ s_{l,x}, s_{l,y} \right]^T$ are:

$$s_{l,y}(p) = \frac{1}{2} \int_{-c_u}^{c_u} S^2(x) - c_v^2 dx, \tag{19}$$

which by applying the mean value theorem becomes:

$$s_{l,y}(p) = \frac{1}{2} \left[ \left( S^2(\chi) - c_v^2 \right) \right] 2c_u, \tag{20}$$

where $\chi \in [-c_u, c_u]$, and

$$s_{l,x}(p) = \frac{1}{2} \int_{S(-c_u)}^{S(c_u)} c_u^2 - \left( S^{-1}(y) \right)^2 dy, \tag{21}$$

which by applying the mean value theorem becomes:

$$s_{l,x}(p) = \frac{1}{2} \left[ c_u^2 - \left( S^{-1}(\xi) \right)^2 \right] [S(c_u) - S(-c_u)], \tag{22}$$

where $\xi \in [S(-c_u), S(c_u)]$. Putting all of the above together, we get:

$$\nabla_p A^T s_l(p) = \tfrac{1}{2}[S(c_u) - S(-c_u)]^2 \left[ c_u^2 - \left(S^{-1}(\xi)\right)^2 \right] + \tfrac{1}{2} \left[ c_v^2 - S^2(\chi) \right] 4c_u^2, \quad (23)$$

which, since $S^{-1} : [-c_v, c_v] \to [-c_u, c_u]$ and $S : [-c_u, c_u] \to [-c_v, c_v]$, is positive as a sum of positive terms. Furthermore, this quantity is zero only when the volume defined by the bounds $[-c_u, c_u]$, $[-c_v, c_v]$ and $S(x)$ is zero. This means that the vectors $\nabla_p A$ and $\dot{p}$ are contradirectional, since $\nabla_p A$ and $s_l(p)$ are codirectional. Thus, $\dot{\mathcal{L}} < 0$, and the system (9) under the control law (10) is asymptotically stable under the proposed assumptions. This concludes the proof. □

An example of a Lyapunov function used in the above proof is depicted in Figure 9.

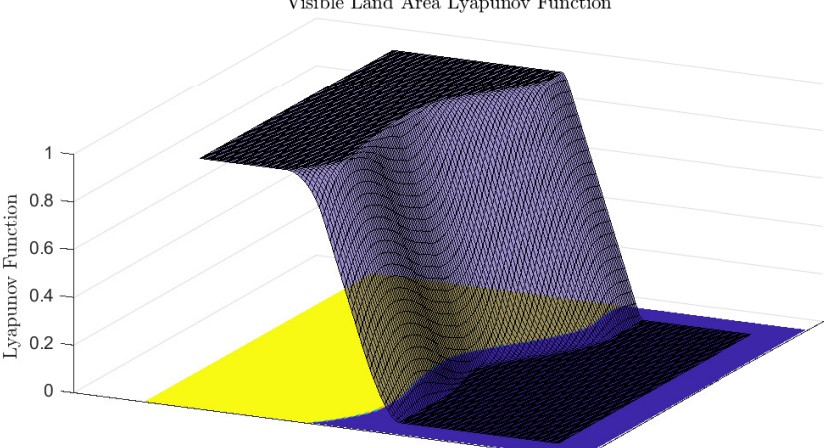

**Figure 9.** Example of a Lyapunov function of Theorem 1.

Evidently, the drone will converge to any part of the state space where the Lyapunov function is equal to zero, which might lead to it drifting indefinitely away from the shore due to external disturbances. This will be avoided through the high-level controller, which will stop the execution. One can visualize the drone's motion under the proposed control law as rolling down the hill of the Lyapunov function. It becomes thus evident that the visible water will be maximized. Note that this analogy is not exact, as the drone will not follow exactly the negated gradient of the Lyapunov function; however, we have shown that it will always move toward the same direction as the aforementioned gradient.

*4.3. Autonomous Landing Module*

4.3.1. Landing Area Detection Algorithm

Having presented the parts of the framework that address the execution of the main mission, we now address the landing problem. For an UAV to land autonomously, while a 2D point might be given by a pilot, the vehicle should examine the topology of the area surrounding the aforementioned point and decide where to land based on the given topology. We assume that prior to starting the landing procedure, the operators of the drone's ground control station have indicated an area for the drone to inspect for landing in the form of a way-point. This area will most likely be in their near vicinity and additionally close to the drone's take-off area (being near the ground control station). Therefore, we omit geological considerations for developing the autonomous landing module and focus primarily on the geometry of the landing spot.

In order to accomplish the above, first, an area of interest around the provided way-point is selected. Then, the latter is discretized into a grid of cells corresponding to points on the ground. In order to determine how fit a specific area is for landing, an appropriate score is formulated where the cells with higher scores are considered more fit for landing.

In this way, an appropriately landing area (according to the drone's footprint) is defined on the fly.

To accomplish this, height measurements need to be acquired in real time. This is achieved by exploiting the Depth Image (Figure 10a) provided by the on-board stereo-camera, but it can be obtained through any other method. Once a Depth Image $D_n(p_x, p_y)$ (where the tuple $(p_x, p_y)$ denotes the drone's position in an inertial frame of reference and the index $n$ is an instant of measurements) is obtained, a procedure of post-processing is followed.

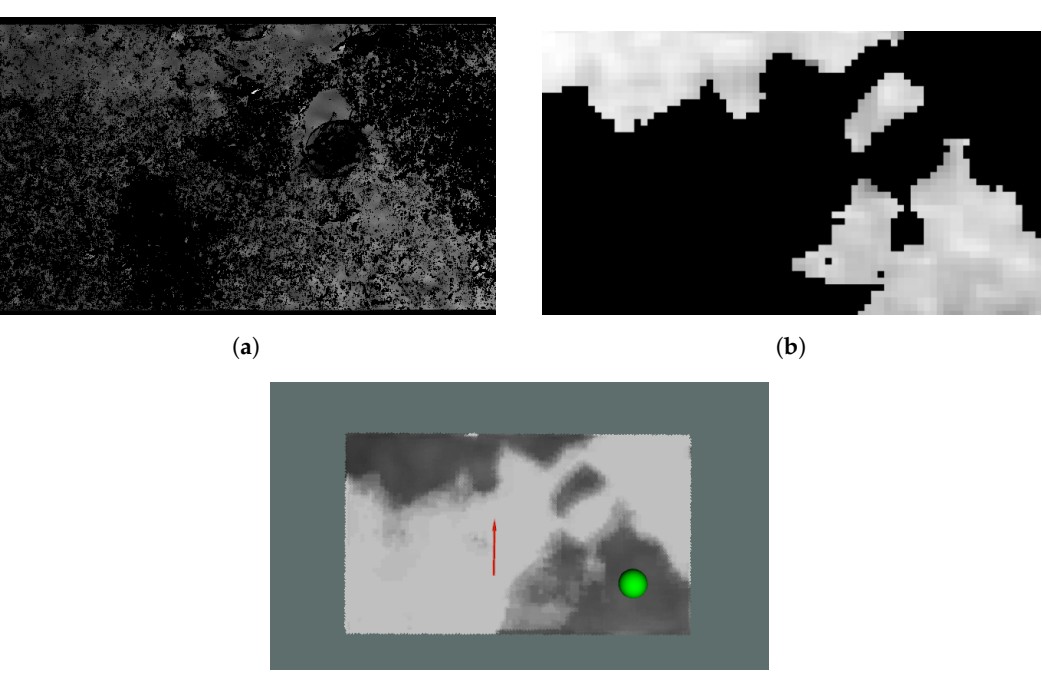

(a)  (b)

(c)

**Figure 10.** (**a**) Depth Image. (**b**) Score Image. (**c**) Occupancy Grid, the best landing spot (marked as green sphere) and the pose of the vehicle (red arrow).

More precisely, the Depth Image is a 2D matrix with dimensions $W \times H$, where $W$ and $H$ are the width and the height of the image. At each pixel $(u, v)$, where $u = 0, \cdots, W - 1$ and $v = 0, \cdots, H - 1$, a distance value $z(u, v)$, expressed in meters, is stored. It should be noted that some elements of the matrix may be characterized as $+\infty$ or $-\infty$ in case that the objects of the corresponding pixels are respectively too far or too close to the camera. Additionally, due to visual occlusions, the estimation of depth may be infeasible or highly inaccurate and, hence, the values of pixels with low confidence are marked as Not a Number (NaN).

In order to evaluate the appropriateness of a pixel for landing, the neighborhood around the query pixel $(u, v)$ is examined. Specifically, a window of size $(K + 1) \times (K + 1)$ is utilized so as to determine the region of interest (ROI) around each pixel $(u, v)$. It is mentioned that the size of the window is highly related to the size of the UAV and is therefore chosen such that the cost of each pixel reflects the ability of the drone to land in the respective surrounding area. The suitability of the surface is quantified by computing the standard deviation $\sigma(u, v)$ of the z-coordinates of the $(K + 1) \cdot (K + 1)$ pixels which constitute the aforementioned region of interest:

$$\bar{z}(u, v) = \frac{1}{(K+1) \cdot (K+1)} \sum_{i=u-\frac{K}{2}}^{u+\frac{K}{2}} \sum_{j=v-\frac{K}{2}}^{v+\frac{K}{2}} z(i, j) \qquad (24)$$

$$\sigma(u,v) = \sqrt{\frac{1}{(K+1)\cdot(K+1)}\sum_{i=u-\frac{K}{2}}^{u+\frac{K}{2}}\sum_{j=v-\frac{K}{2}}^{v+\frac{K}{2}}(z(i,j)-\bar{z}(u,v))^2} \tag{25}$$

As for the pixels that lie next to the borders of the image, a window of smaller size is considered. Additionally, the percentage $\pi(u,v)$ of finite distance values, i.e., values which are not marked as $+\infty$, $-\infty$ or NaNs, is computed inside the region of interest in order to check the validity of the depth information. Regions of interest with percentage $\pi(u,v)$ less than a threshold value $\pi_{min}$ are discarded.

The related score of the query pixel is eventually normalized to $[0,1]$, where score 0 indicates inappropriate areas for landing while 1 indicates appropriate ones, according to the following equation:

$$C(u,v) = \begin{cases} e^{-\sigma(u,v)} & \pi(u,v) > \pi_{min} \\ \\ 0 & \pi(u,v) \le \pi_{min} \end{cases} \tag{26}$$

After post-processing all the pixels of the Depth Image, a new image, namely Score Image (Figure 10b), $C_n(p_x, p_y)$ is constructed, where the score values are stored at each time instant $n$, and a bilateral filter is then applied so as to smooth the Score Image while preserving edges. In order to globally store the associated information, the above scores are matched to the corresponding cells of the Occupancy Grid (Figure 10c), which is expressed with respect to the inertial frame. The matching is performed by exploiting the intrinsic camera parameters, particularly the principal point $c_u, c_v$ and the focal lengths $f_x$, $f_y$ in the $u$ and $v$ directions, respectively, and the current position ${}^I\mathbf{p}_n$ and orientation $\phi_n, \theta_n, \psi_n$ of the UAV. The aforementioned scores are ultimately averaged over the whole set of measurements for each cell:

$$C_n^{avg}(c_x,c_y) = C_{n-1}^{avg}(c_x,c_y)\frac{N_n(c_x,c_y)-1}{N_n(c_x,c_y)} + C_n(c_x,c_y)\frac{1}{N_n(c_x,c_y)} \tag{27}$$

where the tuple $c_x, c_y$ denotes the respective cell of the pixel $u, v$; $N_n(c_x, c_y)$ is the number of costs computed through the respective observations up to the $n$-th measurement instant for the above cell and $C_n(c_x, c_y)$ denotes the score for the respective cell obtained at the $n$-th measurement instant.

Finally, the drone selects the best area to land by finding the cell of the grid that has the maximum score and matching it to a physical 2D position:

$$p_{landing} = p(c_x^l, c_y^l), \tag{28}$$

where

$$(c_x^l, c_y^l) = \arg\{\max_{c_x, c_y}\{C(c_x, c_y)\}\} \tag{29}$$

### 4.3.2. Landing Execution Algorithm

As far as the execution of the landing maneuver is concerned, a model predictive controller (MPC) is formulated in order to complete safely the landing procedure. More precisely, the objective of the control scheme is to minimize the error in position between the vehicle and the aforementioned detected landing location while, simultaneously, satisfying the constraints imposed by the vehicle's low-level velocity controller. The translational kinematics of the multirotor is described by the equation:

$$\begin{bmatrix} \dot{p}_x \\ \dot{p}_y \\ \dot{p}_z \end{bmatrix} = \begin{bmatrix} cos\psi & -sin\psi & 0 \\ sin\psi & cos\psi & 0 \\ 0 & 0 & 1 \end{bmatrix}\begin{bmatrix} u \\ v \\ w \end{bmatrix} \Rightarrow \dot{\mathbf{p}} = \mathbf{R}_z(\psi)^B\mathbf{v} \tag{30}$$

where $\mathbf{p} = \begin{bmatrix} p_x & p_y & p_z \end{bmatrix}^T \in \mathbb{R}^3$ is the position of the vehicle, $^B\mathbf{v} = \begin{bmatrix} u & v & w \end{bmatrix}^T \in \mathbb{R}^3$ is the velocity control input with respect to the body frame of the vehicle, $\psi$ is the yaw angle and $\mathbf{R}_z(\psi)$ is the rotation matrix around the z-axis of the inertial frame.

At each time instant $t$, a constrained optimization problem is solved by the MPC over a finite horizon of $N$ steps and, consequently, an optimal sequence of feasible control inputs $\left( {}^B v_t^*, \cdots, {}^B v_{t+N}^* \right)$ is derived, which minimizes the following weighted sum of accumulative and terminal costs:

$$\min_{^B v_t, \cdots, ^B v_{t+N}} \sum_{k=0}^{N-1} \left( \|p_{t+k} - p_{des}\|_Q^2 + \|^B v_{t+k}\|_R^2 \right) + \|p_{t+N} - p_{des}\|_P^2$$

$$\text{subject to}: p_{t+k+1} = p_{t+k} + R_z(\psi_{t+k})^B v_{t+k} \cdot dt, k = 0, \cdots, N-1$$

$$p_t = p(t)$$

$$^B v_t \in U = \left\{ \forall^B v \in \mathbb{R}^3 : \begin{bmatrix} u_{min} \\ v_{min} \\ w_{min} \end{bmatrix} \leq {}^B v \leq \begin{bmatrix} u_{max} \\ v_{max} \\ w_{max} \end{bmatrix} \right\}$$

where $dt$ is the sample time, $p_{des}$ is the detected landing position and $Q$, $R$ and $P$ are positive definite matrices which penalize the state error, the input and the terminal state error, respectively. According to the receding horizon control principle, only the first element $^B v_t^*$ of the optimal control sequence is applied to the vehicle, and the optimization procedure is repeated at the next time instant $t + 1$, given the measured position $p_{t+1}$, until the successful landing of the vehicle.

### 4.4. High-Level Planner Module

We are now ready to discuss the function of the HLP in more detail. Having presented the function of each module, the HLP decides which module is activated at each time instance. The modules communicate with the HLP via Boolean flags, thus efficiently and quickly passing on information pertaining to the state of their execution. The detailed operation of the HLP is presented in the following Algorithm 1.

---

**Algorithm 1** High Level Planner Algorithm

---

Initialization: WaypointList, LandingWaypoint;
**for** *waypoint in WaypointList* **do**
    Activate *Autonomous Navigation Module*;
    **wait until** *waypoint is reached*;
    Activate *Autonomous Exploration Module*;
    **wait until** *visible water is maximized*;
**end**
Activate *Autonomous Navigation Module* for LandingWaypoint;
**wait until** *LandingWaypoint is reached*;
Activate *Autonomous Landing Module*;

---

## 5. Results

### 5.1. Simulation

In order to validate the performance of the proposed framework on a first level and to ensure the smooth and efficient transition to real-world experiments, rigorous simulation studies are conducted. The simulator is based on the well-known Gazebo [31] and includes highly realistic environments such as terrain heightmaps and water visual effects, which allow the testing of image processing algorithms. An UAV, integrated with the ArduPilot firmware, is used in the simulation scenario, thus allowing Software in the Loop (SITL) simulations without including actual physical hardware.

During the simulation scenario, a list of way-points is commanded and, afterwards, the High-Level Planner Module is responsible for appropriately activating the remaining ones, i.e., Autonomous Navigation, Exploration and Landing. An overview of the Simulation Environment is depicted in Figure 11, while the performance of the framework is illustrated in the relevant video https://youtu.be/5H2HhRz6Oqg (accessed on 25 May 2022).

Following the validation of the proposed framework in a realistic simulation environment (Gazebo-Ardupilot-SITL), real-world experiments are carried out in order to evaluate the efficacy of each one of the modules, which comprise the overall framework, in field conditions.

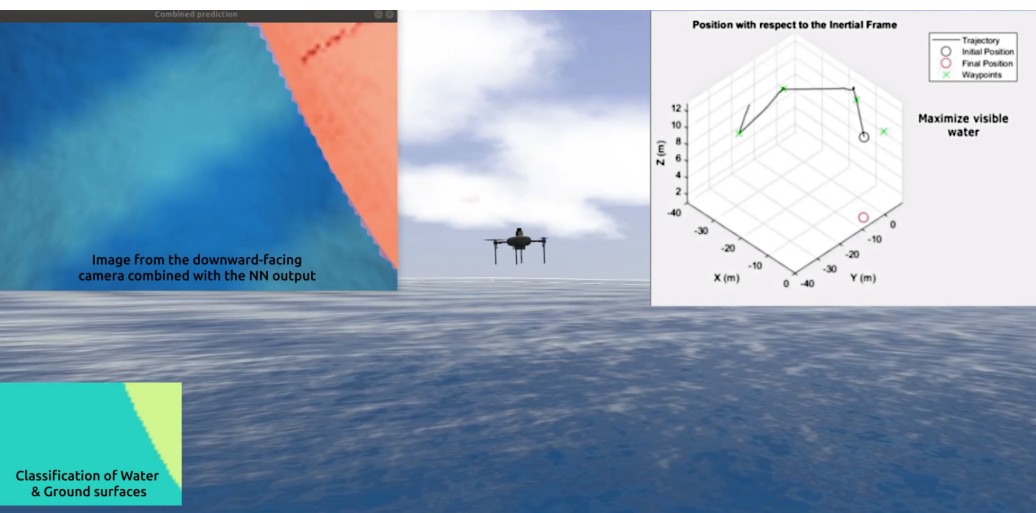

**Figure 11.** Overview of the simulator environment, along with the NN output (pure and overlayed on the camera view) and a 3D plot of the way-points and multi-rotor's trajectory.

### 5.2. Autonomous Navigation Experiments

During a water environmental emergency scenario, a UAV is utilized in order to either realize water sampling or obtain visual feedback for real time or post evaluation of the contaminated aquatic area. Hence, the First Responders command a mission by inserting a list of points, which the UAV should visit, into a common Ground Control Station, and the Autonomous Navigation module is responsible for navigating the vehicle to the desired positions while simultaneously avoiding unexpected and previously unknown obstacles. Although the space above the water level is obstacle-free in most situations, the module is tested in challenging environments where the presence of multiple and dense obstacles jeopardizes the safe completion of the operation.

The first experiment is realized in an outdoor area (Figure 12a) where various obstacles such as rows of trees and street light poles exist and impose difficulties on the safe execution of the mission. A mission, consisting of *four* way-points, is planned, using the Mission Planner [32] Ground Control Station (Figure 12b), in such a manner that the UAV encounters obstacles during its transition from one desired position to another. By exploiting the depth measurements obtained by the forward-looking stereocamera, previously unseen obstacles are detected and a local costmap is constructed in real time (Figure 12c). The local planner continuously produces suitable velocity commands which restrict the navigation of the vehicle to the safe areas of the costmap and, eventually, the target mission is executed without any undesirable collision, as depicted in Figure 12d.

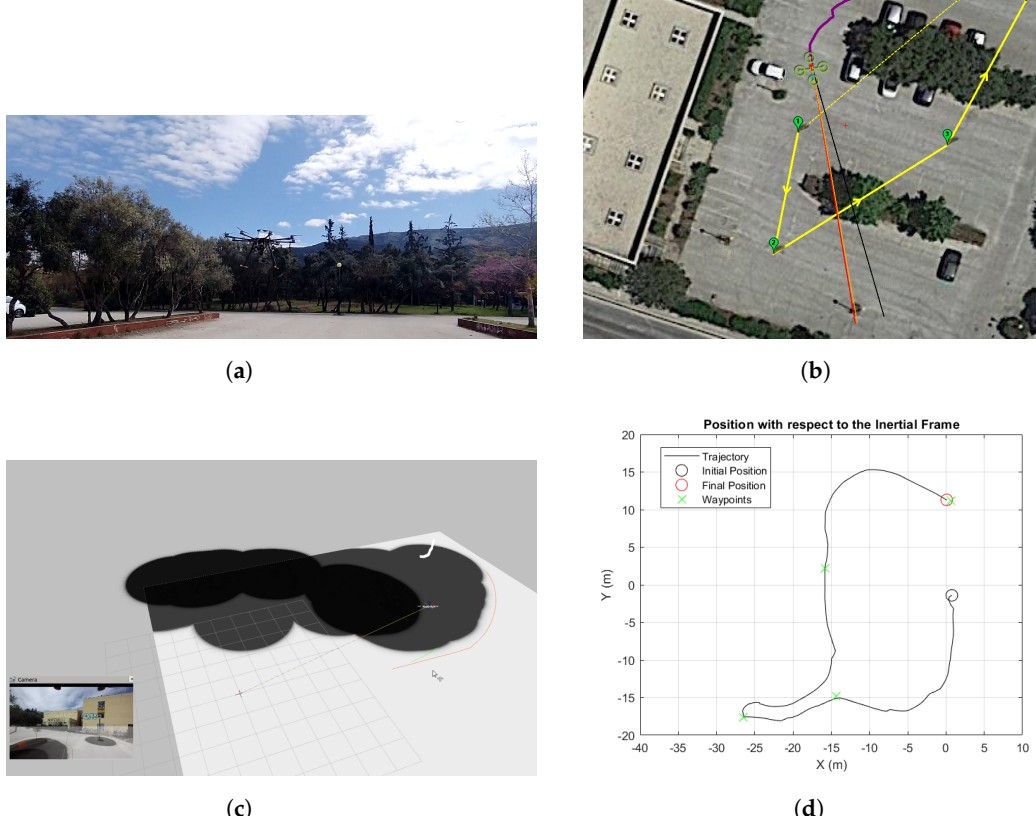

**Figure 12.** An overall view of the Autonomous Navigation Experiment I. (**a**) The environment where Navigation Experiment I is conducted. (**b**) The desired mission of Navigation Experiment I depicted in the Ground Control Station. (**c**) The costmap which is constructed in real time according to the measurements of the stereocamera during the Navigation Experiment I. (**d**) The 2D trajectory of the vehicle and the corresponding desired waypoints of Navigation Experiment I.

The second experiment is more realistic, compared to water contamination emergency situations, since the vehicle flies at a higher altitude and should avoid a high dense tree, which is frequently found within the vicinity of aquatic environments, e.g., lakes and rivers (Figure 13a). In order to evaluate the performance of the Autonomous Navigation module, two desired locations are selected with the obstacle lying between them, as illustrated in Figure 13b. Owing to the existence of an on-board stereocamera, the obstacle is successfully detected and inserted into the costmap (Figure 13c). Re-planning is performed in real time so as to guarantee the safe completion of the navigation task, as shown in Figure 13d.

Both experiments are depicted in the following videos https://youtu.be/FN1uv-WXDho (accessed on 25 May 2022), https://youtu.be/JaP5kiNRwRA (accessed on 25 May 2022).

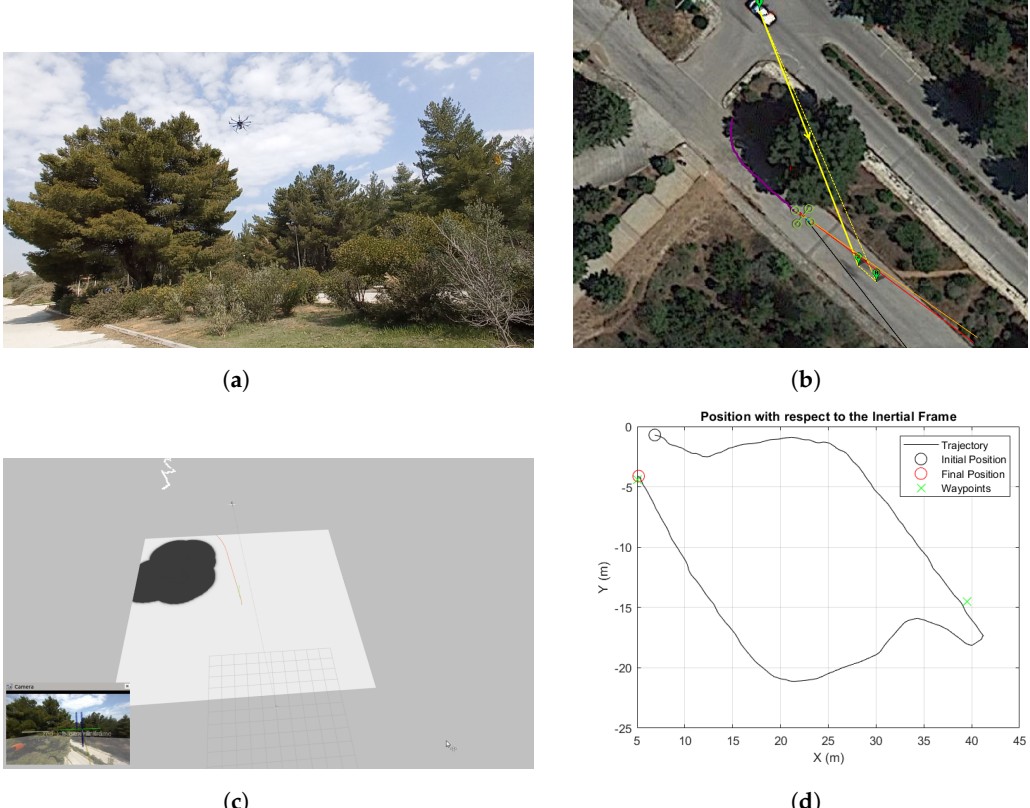

**Figure 13.** An overall view of the Autonomous Navigation Experiment II. (**a**) The environment where Navigation Experiment II is conducted. (**b**) The desired mission of Navigation Experiment II depicted in the Ground Control Station. (**c**) The costmap which is constructed in real time according to the measurements of the stereocamera during Navigation Experiment II. (**d**) The 2D trajectory of the vehicle and the corresponding desired waypoints of Navigation Experiment II.

*5.3. Autonomous Exploration Experiments*

Consider the case in which the First Responders select a way-point in the Ground Control Station in order to execute a sampling procedure or perform water monitoring, and the UAV navigates toward this desired position with the Autonomous Navigation module. However, due to mismatches between the map provided by the Mission Planner and the current actual environment, the image obtained by the on-board downward-looking camera contains both water and ground surfaces. Since the aim of the mission is the sampling or the monitoring of as large a water area as possible, the Autonomous Exploration module is deployed, which is responsible for moving the vehicle until only water is detected.

In order to evaluate the real-time classification of ground and water surfaces, using CNNs, and the capability of the controller to maximize visible water, two experiments are carried out with the orientation, i.e., the yaw angle, of the UAV being initialized arbitrarily. Despite the reflection of sunlight and the existence of rocks and sea waves, the detection of the water surface is achieved uninterruptedly with the CNNs (Figures 14a and 15a). Regarding the performance of the controller, the time evolution of the percentage of "water pixels" in the image plane is depicted in Figures 14b and 15b. The controller exhibits robustness to the waves, due to which a non-monotonic evolution of the percentage is observed, and, eventually, the latter converges to 100% within a few seconds. In both cases, the movement of the vehicle is perpendicular to the wave front regardless of the initial orientation.

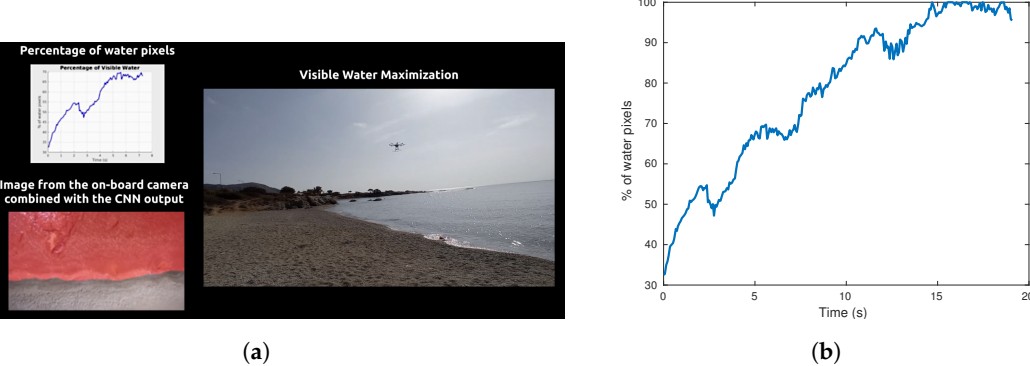

**Figure 14.** Autonomous Exploration Experiment I. (**a**) An overall view of the experiment. (**b**) Time evolution of the percentage of "water pixels".

The performance of this module is better illustrated in the relevant videos https://youtu.be/xn2X1tm9yKk (accessed on 25 May 2022), https://youtu.be/zPOg5yJulTg (accessed on 25 May 2022).

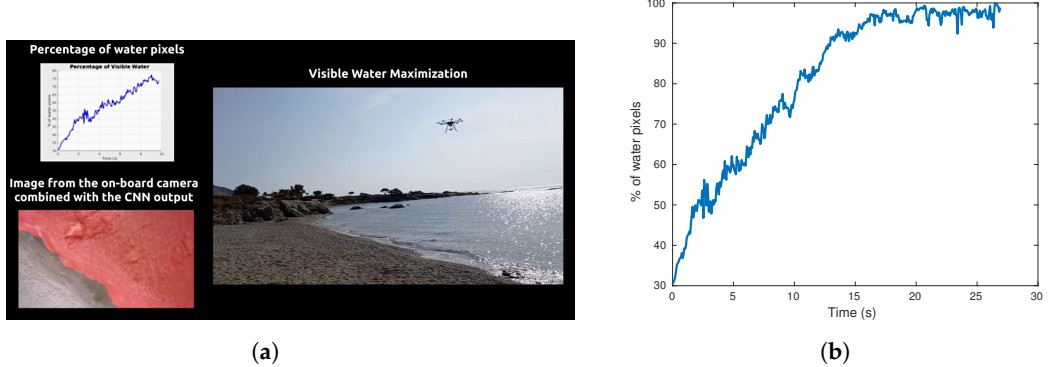

**Figure 15.** Autonomous Exploration Experiment II. (**a**) An overall view of the experiment. (**b**) Time evolution of the percentage of "water pixels".

*5.4. Autonomous Landing Experiments*

Since an autonomous solution is presented, the landing procedure is incorporated into the overall framework and, hence, time and human resources are allocated to other crucial actions during a water contamination emergency situation. After the completion of the commanded mission, the Autonomous Landing module is employed in order to ensure the safe fulfillment of the operation and, hence, the collection of significant data, e.g., water samples, images or video files. By processing the depth measurements of the downward-looking stereocamera, an appropriate area is detected above the final waypoint of the mission and, subsequently, a Model Predictive Controller is deployed in order to efficiently drive the vehicle toward the detected location.

The performance of the Autonomous Landing module is examined in two outdoor environments where the vehicle should autonomously land in the presence of sparse obstacles, vegetation and inclined surfaces, as illustrated in Figures 16a and 17a. In both cases, an appropriate landing spot is identified based on the respective cost maps, which are built in real time (Figures 16b and 17b). The response of the MPC scheme and the evolution of the MPC cost function with respect to time are depicted in Figures 18 and 19. It is evident that the performance of the controller is satisfactory and, hence, the landing maneuver is executed successfully during both experiments.

The experiments are also presented in the following videos https://youtu.be/ZVU_mZ6rZYY (accessed on 25 May 2022), https://youtu.be/_d4rWiSVFug (accessed on 25 May 2022).

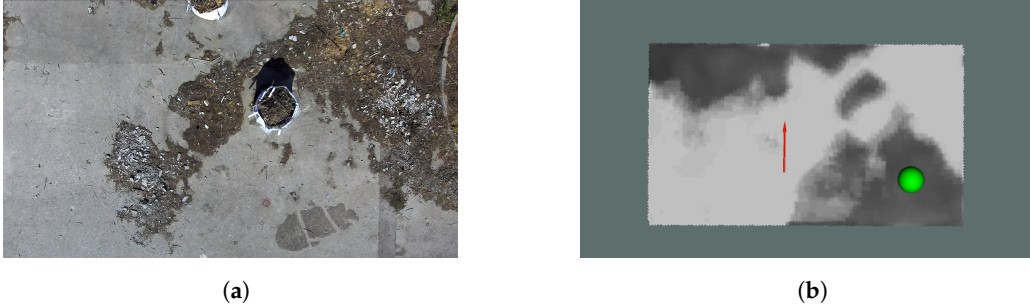

**Figure 16.** The outdoor environment of the Autonomous Landing Experiment I and the respective costmap. (**a**) Image captured by the on-board camera. (**b**) Occupancy Grid, built in real time, the best landing spot (marked as green sphere) and the pose of the vehicle (red arrow).

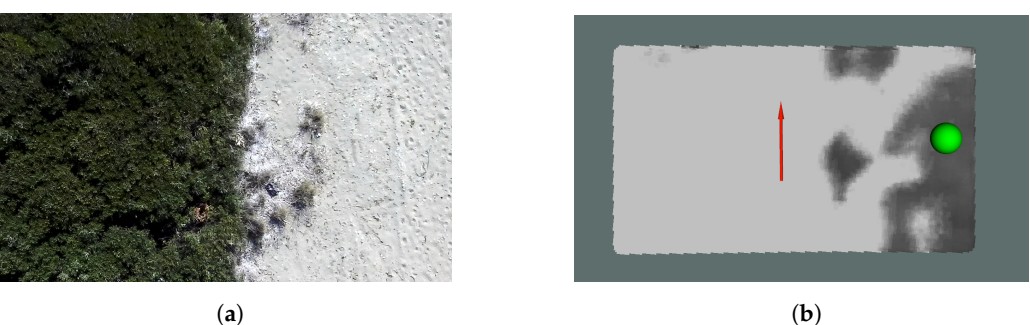

**Figure 17.** The outdoor environment of the Autonomous Landing Experiment II and the respective costmap. (**a**) Image captured by the on-board camera. (**b**) Occupancy Grid, built in real time, the best landing spot (marked as green sphere) and the pose of the vehicle (red arrow).

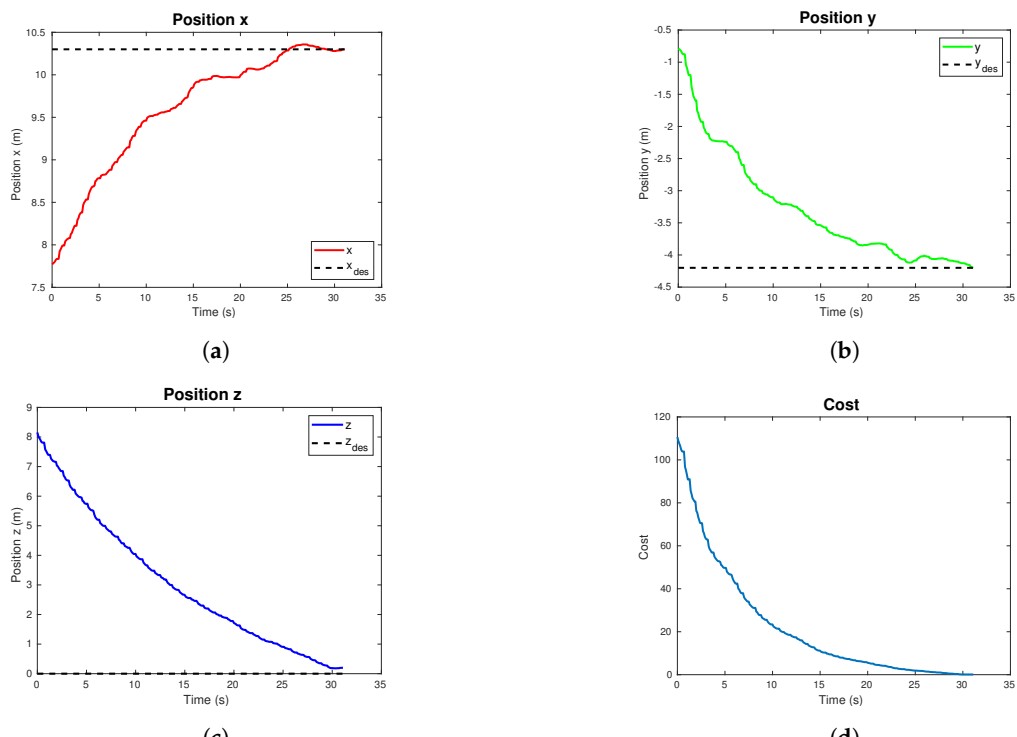

**Figure 18.** The position of the vehicle with respect to the inertial frame, compared to the desired landing spot, and the respective MPC cost function during Autonomous Landing Experiment I. (**a**) Position x. (**b**) Position y. (**c**) Position z. (**d**) MPC Cost Function.

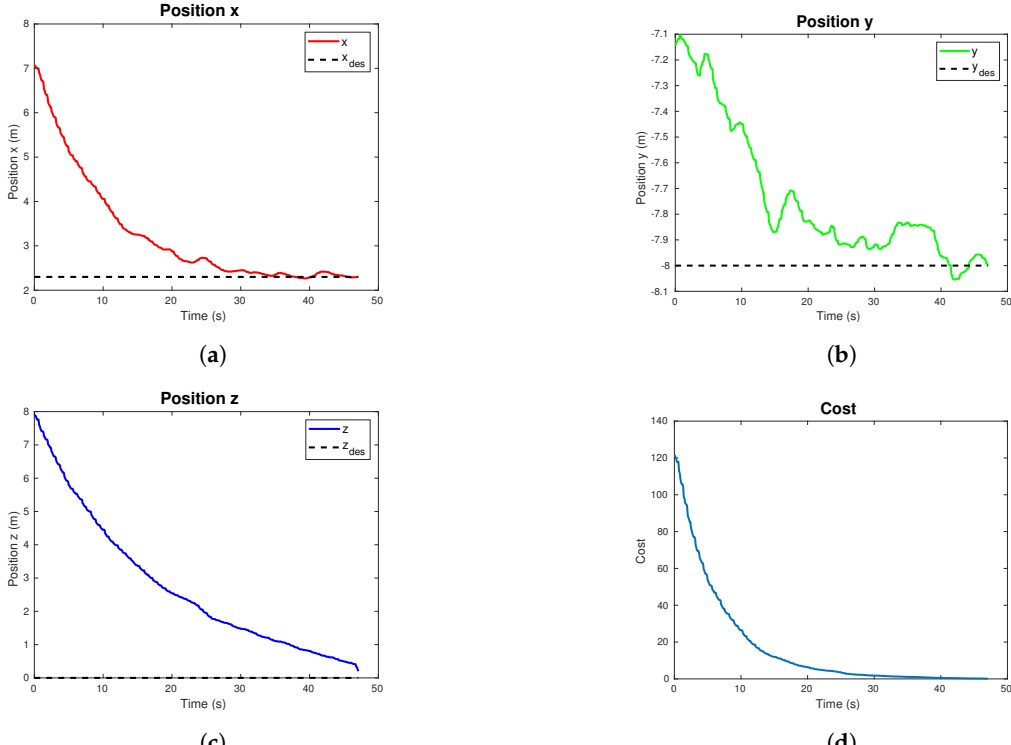

**Figure 19.** The position of the vehicle with respect to the inertial frame, compared to the desired landing spot, and the respective MPC cost function during Autonomous Landing Experiment II. (**a**) Position x. (**b**) Position y. (**c**) Position z. (**d**) MPC Cost Function.

## 6. Conclusions

We have presented a complete, integrated motion planning and control framework that is based on visual feedback for efficiently and effectively monitoring and surveying bodies of water using a UAV. Our framework was developed by combining existing robust solutions in the state-of-the-art of the robotics field, with novel control architectures, that were introduced where necessary, in order to fulfill the goals set out in the problem description. We conclude that after rigorous simulations and real-world field experiments, our framework completes the UAV's mission objectives successfully. The modules presented herein can be employed either as a complete framework or independently, thus providing a modular integrated scheme that can be utilized for various water-related tasks. Additionally, with minor modifications, the presented framework can be employed for other monitoring scenarios, besides water-related ones.

Concerning the water monitoring problem, a very important aspect relates to the acquisition of water samples from an aquatic area of interest. Future promising directions include the integration of various measuring devices on the platform along with the relevant necessary control algorithms in order to expand the capabilities of the proposed solution. More specifically, we intend on incorporating a cable-suspended water sampling mechanism into the UAV's sensor suite and on developing suitable position control algorithms so as to achieve the precise stabilization of the vehicle above the sampling location in the presence of environmental disturbances induced by the water flow.

Finally, an aspect of the landing module that we intend to improve upon relates to considering the geology of the region of interest that the drone examines in order to find an appropriate landing spot. This can be treated through additional classes added to the CNN (which is employed for water detection in this work) such that different soil types can be visually identified and assessed so as to be taken into account during the decision process.

**Author Contributions:** Formal analysis, P.R. and C.B.; Methodology, F.P., P.R. and G.K.; Supervision, G.K., C.B. and K.J.K.; Validation, F.P.; Writing—original draft, F.P. and P.R.; Writing—review & editing, G.K., C.B. and K.J.K. All authors have read and agreed to the published version of the manuscript.

**Funding:** This research was funded from the European Union's Horizon 2020 research and innovation program PathoCERT: Pathogen Contamination Emergency Response Technologies, under grant agreement No. 883484.

**Institutional Review Board Statement:** Not applicable.

**Informed Consent Statement:** Not applicable.

**Data Availability Statement:** No publicly available data were used or generated.

**Conflicts of Interest:** The authors declare no conflict of interest.

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
