# Peer review of "A Vision-Based Motion Control Framework for Water Quality Monitoring Using an Unmanned Aerial Vehicle"

_sustainability, doi:10.3390/su14116502_

Round 1

Reviewer 1 Report

The authors have proposed an end-to-end solution for the safe and autonomous navigation of the UAV. However, several questions were raised:

  1. Why did the authors focus on Water Quality monitoring?
  2. Can the same method be used for monitoring other objects, such as plants? what is your novelty then? please explain
  3. Why did the authors not collect water quality data and calibrate it with laboratory analysis? this method is compulsory for estimating water quality through UAV sensor
  4. I recommend to the authors to revise the whole manuscript

Author Response

REVIEWER 1

Comments and Suggestions for Authors

The authors have proposed an end-to-end solution for the safe and autonomous navigation of the UAV. However, several questions were raised:

  1. Why did the authors focus on Water Quality monitoring?
  2. Can the same method be used for monitoring other objects, such as plants? What is your novelty then? please explain
  3. Why did the authors not collect water quality data and calibrate it with laboratory analysis? this method is compulsory for estimating water quality through UAV sensor
  4. I recommend to the authors to revise the whole manuscript

RESPONSE TO REVIEWER 1

We thank the Reviewer for the comments. 

  1. We would like to highlight that the proposed modular framework can be extended to address a plurality of scenarios. In this paper we have however focused on monitoring of water resources, as the current Special Issue explicitly concentrates on water quality monitoring (for further details we provide the relevant information: Special Issue Information). 
  2. As discussed previously, the herein presented modular framework can indeed fit other problem formulations, given the appropriate changes are made to some (or all of the) modules. For instance, to address a plant-monitoring problem, the CNN employed in our original work for water detection could be re-trained to classify any relevant plant aspect. The novelty of our work persists; namely the modular nature of the framework and the integration of the modules to a unified framework (see Subsection 1.2, final paragraph).
  3. We thank the Reviewer for the comment. Indeed, water sampling is a pivotal aspect of water quality monitoring. Nevertheless, this step can be easily integrated within our framework, through the addition of an extra module. In future works, we aim at developing and incorporating such a control/software module along with a relevant water-sampling mechanism. Up to this point, we place greater emphasis on visual water monitoring, which in itself is a very important tool for assessing water quality with respect to some metrics. We have altered the manuscript accordingly to reflect the herein discussed points, as well as our intended future work (see Section 6). 
  4. We thank the Reviewer for the comment. We have revised the manuscript according to the Reviewers’ points.

Reviewer 2 Report

1.In autonomous exploration module,there is no enough evidence that the CNN water detection network is robust  in foggy  or dusky or other unfavorable conditions. Please provide more details.

2.In autonomous landing module,it is not rigorous to just consider the standard deviation.Maybe you should take into account the hardness and soil type of landing area.

Author Response

REVIEWER 2:

Comments and Suggestions for Authors

  1. In the autonomous exploration module, there is not enough evidence that the CNN water detection network is robust in foggy or dusky or other unfavorable conditions. Please provide more details.
  2. In the autonomous landing module, it is not rigorous to just consider the standard deviation. Maybe you should take into account the hardness and soil type of landing area.

RESPONSE TO REVIEWER 2

We thank the reviewer for the constructive comments. 

  1. We agree with the Reviewer on the point that the generalization capabilities of Neural Networks is a crucial aspect in the context of field robotics. However, this work aims at initially creating a modular interconnected framework for multi-rotor applications, while improving the capabilities of each module is an important step that we intend to expand upon in future works. With respect to the Reviewer’s point, this includes rigorous data collection from a variety of environmental conditions and the re-training of the CNN. We have altered the related subsection of our manuscript to include this discussion (see Subsection 4.2.1).
  2. As the Reviewer has astutely noted, the standard deviation only provides information about the geometry and the topology of the candidate landing area. However, we assume that prior to starting the landing procedure, the operators of the drone’s ground control station have indicated an area for the drone to inspect for landing, in the form of a waypoint. This area will most likely be in their near vicinity and additionally close to the drone’s take-off area (being near the ground control station). Therefore, we assume that the geology of the candidate landing area will be more or less suitable for landing and is thus not a critical factor for the success of the mission. Nevertheless, integrating a classification CNN that accounts for the geology of the landing area is an interesting future direction. We have added this direction to the  “Conclusions” section of our manuscript  (see Section 6). We have furthermore altered Section 4.3 of our manuscript to clarify the herein discussed points  (see Subsection 4.3.1).

Reviewer 3 Report

Dear authors thank you very much for the interesting and valuable text. Wheel described methodology, advanced mathematical explanation, and (what is most important complete video confirmation of performed experiments), make this article complete and extremely interesting. 

Described algorithms for 'landing area detection' and 'visible water maximization algorithm' are interesting approaches and can be transferable also for other UAV implementations. 

The first impression was that the text is a bit too long but during the reading, it turned out that all parts of the text are organized in a logical continuum.

Author Response

REVIEWER 3:

Comments and Suggestions for Authors

Dear authors, thank you very much for the interesting and valuable text. Wheel described methodology, advanced mathematical explanation, and (what is most important complete video confirmation of performed experiments), make this article complete and extremely interesting.

Described algorithms for 'landing area detection' and 'visible water maximization algorithm' are interesting approaches and can be transferable also for other UAV implementations.

The first impression was that the text is a bit too long but during the reading, it turned out that all parts of the text are organized in a logical continuum.

RESPONSE TO REVIEWER 3

We thank the Reviewer for his/her comments, and for the fair evaluation of our work. We have indeed developed a modular framework that can be employed and/or modified to fit several applications. We also find that the plurality of modules necessitates the extensive text and we therefore thank the Reviewer for commending the emphasis given to the structure of the submitted text.

Reviewer 4 Report

The manuscript have presented a complete, integrated motion planning and control framework that is based on visual feedback for efficiently and effectively monitoring and surveying bodies of water using an UAV.

Author Response

REVIEWER 4:

The manuscript presents a complete, integrated motion planning and control framework that is based on visual feedback for efficiently and effectively monitoring and surveying bodies of water using an UAV. 

RESPONSE TO REVIEWER 4

We thank the Reviewer for the fair description and evaluation of our work. Our work was significantly improved by integrating all of the Reviewers’ comments.

Round 2

Reviewer 1 Report

I recommend accepting this present form of manuscript